# Enhanced Collaborative Filtering for Personalized E-Government Recommendation

**Ninghua Sun** [1,2], **Tao Chen** [1,*], **Wenshan Guo** [1] **and Longya Ran** [1]

[1]  School of Public Administration, Huazhong University of Science and Technology, Wuhan 430074, China; ninghua_sun@hust.edu.cn (N.S.); wenshanguo@hust.edu.cn (W.G.); d201881080@hust.edu.cn (L.R.)
[2]  Innovation Institute, Huazhong University of Science and Technology, Wuhan 430074, China
[*]  Correspondence: egov@hust.edu.cn

**Abstract:** The problems with the information overload of e-government websites have been a big obstacle for users to make decisions. One promising approach to solve this problem is to deploy an intelligent recommendation system on e-government platforms. Collaborative filtering (CF) has shown its superiority by characterizing both items and users by the latent features inferred from the user–item interaction matrix. A fundamental challenge is to enhance the expression of the user or/and item embedding latent features from the implicit feedback. This problem negatively affected the performance of the recommendation system in e-government. In this paper, we firstly propose to learn positive items' latent features by leveraging both the negative item information and the original embedding features. We present the negative items mixed collaborative filtering (NMCF) method to enhance the CF-based recommender system. Such mixing information is beneficial for extending the expressiveness of the latent features. Comprehensive experimentation on a real-world e-government dataset showed that our approach improved the performance significantly compared with the state-of-the-art baseline algorithms.

**Keywords:** e-government public services; collaborative filtering; recommender system; negative sampling

## 1. Introduction

E-government refers to providing online applications to enhance the access to, and the delivery of, government information and service to the public for citizens by using modern technologies [1,2]. As more and more professional services are loaded on the website, however, it is harder for ordinary citizens without expertise to quickly find the target service from hundreds of service items in a certain scenario of e-government because of information overload [3,4]. The existing solutions for the e-government platforms are personalized recommendations that aim to find out target service items for the users based on their preferences, behaviors, and other information [5].

The collaborative filtering (CF) algorithm is one of the most widely used recommender algorithms since it collectively learns users and items latent representations from the user–item rating matrix [5–7]. In most recommender models, this matrix is binary with a value of 1 representing implicit feedback (e.g., click, browse, and collection) between a user and an item, and a value of 0 otherwise. In the e-government recommendation scenarios, the users' preferences also are implicit feedback—such as document browsing, online certificate status query, and new-born settlement. However, most existing e-government recommendation models need an explicit indication of users' preference (i.e., ratings) which most e-government platforms are unable to provide. Therefore, how to learn users' preferences from implicit feedback based on CF is key to the e-government recommender task.

Advanced CF models focus on the use of embedding technology to obtain users and items latent features, such as matrix factorization (MF) [5,6] and neural networks [8–10].

Neural collaborative filtering [6] (NCF) is a popular pointwise CF algorithm for recommendation with implicit feedback, which leverages the flexibility and non-linearity of neural networks to replace dot products of matrix factorization. In specific, this model is structured with two subnetworks including generalized matrix factorization (GMF) and multilayer perceptron (MLP) and models the interactions from two pathways instead of simple dot products. After the NCF, many algorithms have been proposed, which have mainly focused on how to enhance the expressiveness of users and items embedding features with the implicit data [11–13]. However, together with NCF, they have made a premise of user–item interactions: for a user, he disliked un-interacted items. They ignored the effect of negative items on positive items.

In this paper, we argue that this premise may hinder the expression of interacted items and our understanding of user's needs, that is some unrated items should be viewed as the potential preference for the user. Consider the following example: a user queried the tourist flow on the e-government website, and unclicked other services. As shown in Figure 1: intuitively, the user will book a travel appointment in the future, which means the service "appointment" may be the false-negative item. When training the recommender model, such potential preferences may enhance the embedding feature of items and lead to a better result.

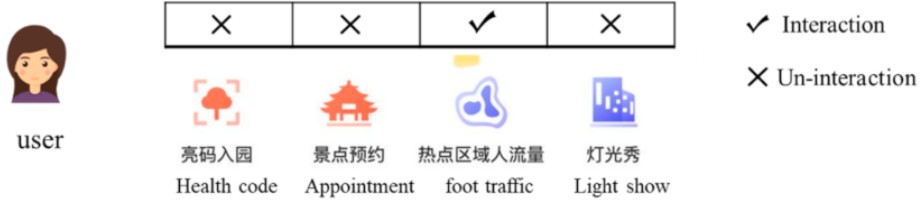

**Figure 1.** Illustration example of items in the e-government website.

This study aims to return a list of recommended items while relaxing the assumption that all the un-interaction items are not the users' preferences in training. In this work, we also treat the implicit feedback as a binary rating matrix. Factually, it is hard to know whether an un-interaction item is true negative or false negative. In our solution, two strategies are designed: negative items sampling and negative mixing. In the negative items sample, we try to construct a function to evaluate a user's preference for the negative items, and then select a better one from them. In negative mixing, we introduce a mixing method to enhance the embedding of positive items by injecting the information from negative items into them. We summarize the main contributions of this thesis as follows:

- We propose a simple negative item mixing collaborative filtering framework (NMCF) to enhance the representation of items. Unlike other neighborhood-based methods, this method mixed the latent features of the negative item with the positive item. The mixed layer effectively simulated the connectivity of the interaction graph and captured the high-order features. To the best of our knowledge, this is the first to introduce the idea of mixing negative information to the personalized e-government service recommendation.
- To obtain more information on user preferences, we proposed an effective time-sensitive distance measure for the recommendation. Unlike other methods, the distance can measure the similarity between the interaction pairs, which is advantaged in describing the interaction data in measure space visually.
- We conduct comprehensive experiments on a real-world e-government dataset, which is provided by the Administrative Examination and Approval Bureau of Wu Hou District, Chengdu. The result demonstrated the effectiveness of our model for the personalized public service recommendation task.

The rest of this article is organized as follows. Section 2 elaborates on the related research. The proposed method is described in Section 3. Section 4 presents an empirical study on the Wuhou services dataset. Finally, Section 5 discusses and concludes our work.

## 2. Related Work

In this section, we provide a brief review of previous works on collaborative filtering algorithms for a personalized recommendation. Collaborative filtering technology can be further classified into two types: neighborhood-based recommendation systems and model-based recommendation systems.

### 2.1. Collaborative Filtering Recommendation

As a typical recommender system technique, the collaborative filtering algorithm has drawn a large amount of attention from academic circles due to its excellent performance in dealing with personalized recommendations. Its success has inspired some efforts at using CF in the modeling of recommendation systems. Especially, refs. [14,15] provided a novel recommendation model based on CF to return the best top-K items for users.

Generally, the feedback information can be divided into two types: explicit and implicit feedback. Explicit feedback, such as rating scales, is a scoring mechanism used to express users' explicit preference over items or services. Implicit feedback, such as browsing or clicking, is automatically collected by the recommender system [16]. Generally, the collaborative recommendation system for dealing with these behaviors is categorized into two directions: neighborhood-based or memory-based collaborative filtering [17,18] and model-based collaborative filtering [19,20].

#### 2.1.1. Neighborhood-Based Recommendation System

There are two different neighborhood-based recommendations based on the k-nearest neighbor algorithm [21]: Item/object-based collaborative filtering and user/customer-based collaborative filtering. Item-based and user-based collaborative filtering techniques (item-based CF and user-based CF) are based on full raw ratings in a user–item matrix. The goal of these technologies is to find similar items or users. The most used similarity metrics include the Pearson correlation coefficient and cosine similarity. However, these approaches are more vulnerable to suffering from the sparse data problem. This fact may cause difficulty in computing similarity and evaluating the rating matrix.

Recently, some scholars attempt to solve this problem. Reference [22] proposed an algorithm that combines item-based CF and deep learning. It not only models the interaction between two items by using similarity metrics but also models the interaction among all interacted item pairs by using neural networks. Reference [23] presented that there is a method to combine item-based CF and user-based CF according to a new similarity measure.

#### 2.1.2. Model-Based Recommendation System

In terms of model-based recommendation algorithms, not only the initial rating data but also the latent parameters of the model are used to make predictions [24]. Several common kinds of model-based recommendation algorithms—e.g., MF and neural networks—are available. MF, as a classical model-based CF, uses the latent features of users and items to make rating predictions. The widely adopted optimization algorithms for MF include adaptive moment estimation [25], stochastic gradient descent [26], and alternating least squares [27]. It usually achieves better performance in comparison to nearest-neighbor techniques.

To improve the accuracy and expressiveness of MF, embedding technology is introduced to learn the latent features. Embedding layer is frequently used in the neural network models, which can effetely transform a high-dimensional sparse binary vector (i.e., one-hot vector) into a dense low-dimensional vector. A simple user embedding construction of the user $u$ which belongs to the user set $U$ is presented in Figure 2. The figure shows that the dimension of the input one-hot vector $x^{(u)}$ is $|U|$. Additionally, the element $x_j^{(u)}$ is in the $j^{th}$, corresponding to the real binary value of the one-hot vector. The output dimension is $m$ and user embedding is expressed in the equation

$$e_u = Wx^{(u)} = w_u$$

where, $W \in R^{m \times |U|}$ describes the weight matrix mapped the user one-hot vectors into lower-dimensional vectors. Analogously, the embedding layer can also be used to get the item latent feature. As a typical neural network learning algorithm, gradient descent, or its variants such as adaptive moment estimation (Adam) is widely adopted to update the weights of the embedding layer.

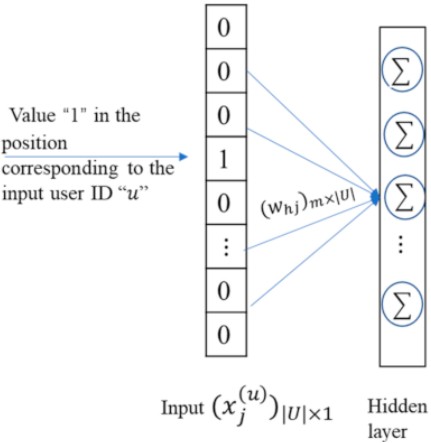

**Figure 2.** Example embedding layer of an input user ID.

### 2.2. Hybrid Recommendation

To better learn the representations of users and items, many current models attempted to propose hybrid algorithms [28,29]. For example, work [30] integrates MF with a marginalized denoising auto-encoder to effectively learn users' and items' latent features. It is also efficient to introduce neighbor-based models in MF [31] and social network-based to extend the process of similarity computation for generic modeling of features [32]. In recent years, neural network models have been integrated into the MF framework to search high-order user or item latent features [33], which have achieved much better performance. Thus, it is natural to explore the feasibility of applying this model to the e-government services recommendation. Additionally, some scholars argued that merely using the rating scores may not be enough to model users' preferences. Therefore, work [34] proposes a hybrid model by introducing auxiliary information, such as users' ratings, reviews, and social data. Work [35] not only learned the rating data but also captured features from reviews data with the help of a convolutional neural network (CNN). Work [36] also uses the reviews information to mining users' potential preferences for the items based on the naive Bayes algorithm.

Compared with the above-mentioned method, the proposed method in our work has such advantages: (1) one of the most basic components of the model is the negative item sampling that can capture the high order indication of users' potential preferences. We extract and then propagate the high-order information to get better predictions. (2) This model has constructed a time-sensitive distance, which provides abundant information to evaluate the correlation of the interactions under limited information.

### 3. Proposed Negative Mixing CF

In this part, we first clarify the notations used in this paper and explain the detailed selection of the negative items for each positive item. Then, we present how the negative item was mixed with the positive item. Finally, we construct the prediction layer for a recommendation.

### 3.1. Notations

In the implicit feedback scenario, we denote the user–item rating matrix $Q \triangleq (y_{ui})_{|U| \times |I|}$, where $U$ and $I$ are the sets of all the users and items, respectively. $y_{ui} = 1$ represents an interaction $(u, i)$, while $y_{ui} = 0$ represents user $u$ has not interacted on item $i$. Naturally, the

implicit feedbacks are denoted as $Q^+ \triangleq \{(u,i)|y_{ui} = 1, u \in U, i \in I\}$, and the set $U^i \subseteq U$ are the users that both interacted with item $i$.

### 3.2. Negative Items Sampling

Recall that the nodes set in the connected graph can be transformed into a metric space by defining the metric function. In this way, the distances can be calculated as the weighted shortest-path distance between two nodes in the graph. In the recommender scenario, the user–item interaction histories can be represented as an undirected graph $G = (V, E)$ where the nodes set $V$ are the union of users set $U$ and items set $I$ and the edges set $E$ each of which is composed of a node pair indicating that the user $u$ interacts with item $i$ (Figure 3). Therefore, we can calculate the distance between the users and items based on the connectivity of the interaction graph.

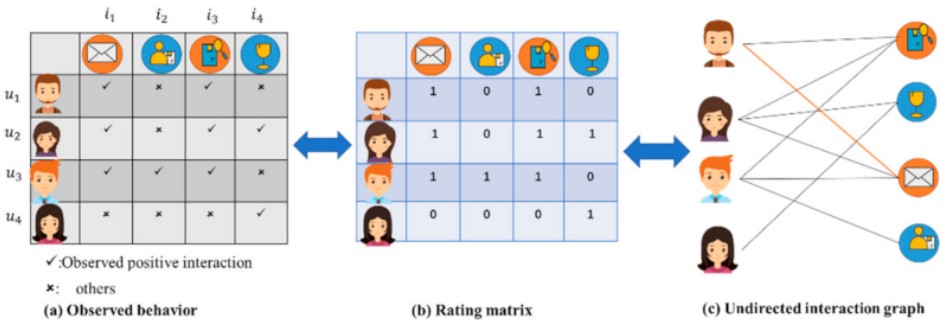

(a) Observed behavior     (b) Rating matrix     (c) Undirected interaction graph

**Figure 3.** Transformation of rating matrix and graph.

We first define the metric function $d_{(u,i)} : E \to R$ (each edge is mapped to a real number) for the graph $G$ based on the auxiliary information (e.g., interaction timestamp) of the feedback and demographic information (e.g., age). Therefore, the weight for $(u, i) \in E$ is

$$d_{(u,i)} = d_{(i,u)} = \left( f_{(u,i)}(\Delta t, n) - C_{ave} \right) \quad (1)$$

$$f_{(u,i)}(\Delta t, n) = \frac{\gamma_2(n - N)^2 + e^{-\gamma_1 \frac{1}{|\Delta t|}}}{2} \quad (2)$$

where $f_{(u,i)}(\Delta t, n)$ evaluates the user $u'$s preference for item $i$. Specifically, $\Delta t$ is the time delay between user $u'$ interaction at timestamp $t$ and the latest interaction, and $n$ is the age when user $u$ first interacted with item $i$, $N$ is the mode of users age in the training set, and $\gamma_1, \gamma_2$ are scale factors. A smaller deviation of age $n$ from $N$ or smaller time delay $\Delta t$ leads to a closer distance between $(u, i)$. $C_{ave}$ is the average of $f_{(u,i)}(\Delta t, n)$ overall interaction pairs. Considering the example: for the interaction path $(u_1 \to i_1 \to u_2 \to i_2)$, the distance between the user $u_1$and its negative item $i_2$ can be formulated as

$$d_{(u_1,i_2)} = d_{(u_1,i_1)} + d_{(i_1,u_2)} + d_{(u_2,i_2)} \quad (3)$$

Then we present a negative sampling strategy based on the defined function. For each active interaction pair $(u, i)$, we select the negative items from the interaction pairs set $H_{(u,i)}$ of which each user is connected to the positive item $i$. Formally,

$$H_{(u,i)} = \left\{ (u', i') \big| y_{u'i} = 1, y_{u'i'} = 1, u' \neq u, i' \neq i \right\} \quad (4)$$

We conform to the following information selection strategy for the learning process:

- For positive interaction $(u, i)$ and $H_{(u,i)} \neq \emptyset$, if the distance value $d_{(u,i')}$ is less than the given constraint value $\varepsilon$, the negative item $i'$ is selected to mix with positive item $i$. The constraint $\varepsilon$ worked as a 'gate' described in Figure 4.

- For positive interaction $(u, i)$, $H_{(u,i)} = \varnothing$, the positive item $i$ is selected to mix with itself.
- For a negative interaction $(u, i)$, the item $i$ is also be selected.

Also, a pseudo code of negative interaction selection for positive interaction is performed in Algorithm 1.

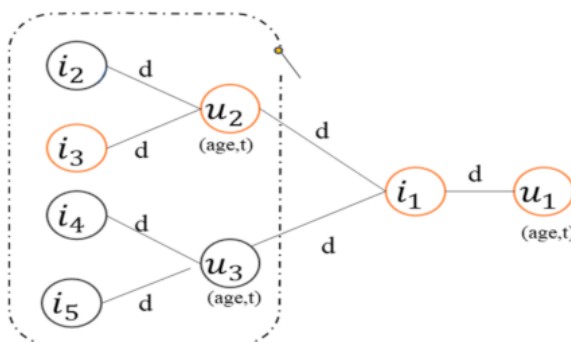

**Figure 4.** Information selection strategy with distance constraint.

### 3.3. Negative Mixing

For an interaction $(u, i)$ and its selected interaction $(u', i') \in H^+_{(u,i)}$, we denote their latent feature vectors as $(e_u, e_i)$ and $(e_{u-}, e_{i-})$, respectively. Inspired by the mixup [37], we introduce the idea of negative mixing to inject negative information $e_{i-}$ into positive embeddings. Specially, we compute the composite item feature $e'_i$ by a linear combination of $e_i$ and its selected negative item feature $e_{i-}$

$$e'_i = e_i + s(e_i, e_{u-})e_{i-} \tag{5}$$

where score $s(e_i, e_{u-})$ is the decay factor of the selected item, which evaluates how much information is propagated from $e_{i-}$ to $e_i$ conditioned to their mutually interactive user latent feature $e_{u-}$. Thus, decay factor $s(e_i, e_{u-})$ should contain the collaborative information between positive item latent vector $e_i$ and user latent vector $e_{u-}$. Generally, the inner product can be more collaboratively informative than the connection. The calculation formula is

$$s(e_i, e_{u-}) = \langle e_i, e_{u-} \rangle \tag{6}$$

The expressiveness of the item latent feature has been enhanced because of the mix of the selected information. Taking $(i_2 \rightarrow u_2 \rightarrow i_1 \rightarrow u_1)$ or $(i_2, u_2) \rightarrow (i_1, u_1)$ as an example, $i_2$ propagates its latent feature information to $i_1$ via $u_2$, which is beneficial for better estimation of $u_1$'s preferences. The interaction $(u_2, i_2) \in H_{(u,i)}$ is selected to provide implied information for positive interaction $(u_1, i_1)$. A high collaborative filtering score of $(u_2, i_1)$ suggests that more latent feature $i_2$ can provide. The process is necessary for the binary rating data of e-government, which loses explicit preference expression. The mixed information introduces a deeper interaction to enhance the abstract expression of connected interactions.

---

**Algorithm 1:** Negative interaction selection for positive interaction.

**Input:** rating matrix $\boldsymbol{Q}$; interaction $(\boldsymbol{u}, \boldsymbol{i})$; threshold $\boldsymbol{\varepsilon}$

**Output:** the selected interaction $(\boldsymbol{u'}, \boldsymbol{i'})$ for $(\boldsymbol{u}, \boldsymbol{i})$

1.     **for** $\boldsymbol{u'}$ in $(\boldsymbol{U})^{(i)} \backslash \{\boldsymbol{u}\}$ **do**
2.         **for** $\boldsymbol{i'}$ in $\boldsymbol{I} \backslash \{\boldsymbol{i}\}$ **do**
3.             **if** $\boldsymbol{y_{u'i'}} = \boldsymbol{1}$ **do**
4.                 add $(\boldsymbol{u'}, \boldsymbol{i'})$ in set $\boldsymbol{H_{(u,i)}}$;
5.             **end if**
6.         **end for**
7.     **end for**
8.     randomly sample $(\boldsymbol{u'}, \boldsymbol{i'})$ from $\boldsymbol{H_{(u,i)}}$;
9.     calculate $\boldsymbol{d_{(u',i')}}, \boldsymbol{d_{(u,i)}}, \boldsymbol{d_{(u',i)}}$ according to Equations (1)–(3);
10.    $\boldsymbol{d = d_{(u,i)} + d_{(u',i)} + d_{(u',i')}}$
11.    **while** $\boldsymbol{d > \varepsilon}$ **do**
12.        randomly sample $(\boldsymbol{u'}, \boldsymbol{i'})$ **from** $\boldsymbol{H_{(u,i)}}$;
13.        $\boldsymbol{d = d_{(u,i)} + d_{(u',i)} + d_{(u',i')}}$
14.    **end while**
15.    **return** $(\boldsymbol{u'}, \boldsymbol{i'})$

---

### 3.4. Prediction Layer

In this subsection, we present the prediction layer based on the original MF predictor. In our work, MF characterizes each item and user by latent features obtained from the embedding layer. For each interaction $(u, i)$, we will estimate a predictor $\hat{y}_{ui} \in (0, 1)$ that evaluates the probability of user $u$ will act on item $i$. The rating factor $\hat{y}_{ui}$ integrated the original predictor $\hat{y}_{ui}^1$ and mix-based predictor $\hat{y}_{u,i_{mix}}^2$ in the equation

$$\hat{y}_{ui} = \sigma\left(w_1 \hat{y}_{ui}^1 + w_2 \hat{y}_{u,i_{mix}}^2 + b\right) \tag{7}$$

where $\sigma$ and $w_i (i = 1, 2)$, are the sigmoid function, and the weights to be trained, respectively. Additionally, $b$ stands for the bias item. We obtain original predictor $\hat{y}_{ui}^1$ and mix-based predictor $\hat{y}_{u,i_{mix}}^2$ by

$$\hat{y}_{ui}^1 = \langle e_u, e_i \rangle \tag{8}$$

$$\hat{y}_{u,i_{mix}}^2 = \langle e_u, e_i' \rangle \tag{9}$$

### 3.5. Optimization

3.5.1. Objective Function

For each interaction pair $(u, i)$, the learning task can be formulated as a supervised binary classification that predicts the probability of user $u$ prefers item $i$. Naturally, we adopt the log loss objective function for evaluating the performance:

$$\mathcal{L} = -\sum_{(u,i) \in Q^+ \cup Q^-} y_{ui} \log \hat{y}_{ui} + (1 - y_{ui}) \log(1 - \hat{y}_{ui}) \tag{10}$$

For the negative cases $Q^-$, we uniformly sample them from unobserved (negative) interactions in each iteration and control the sample ratio. To avoid overfitting, the regularizing term $\| \theta \|_2^2$ was adopted to penalize the magnitudes of the parameters, and the regularizing term $\lambda$ was determined by cross-validation. Finally, the objective function of the model is expressed to learn Equation (10):

$$\mathcal{L}_{ours}(\theta) = \mathcal{L} + \lambda \| \theta \|_2^2 \tag{11}$$

where $\theta = \{W_E, W_{MF}\}$ indicates the weight parameter of the model, $W_{MF} = \{w_1, w_2\}$ is the output layer weights; $W_E$ is the set of embedding layer weights that allow us to get user and item latent features flexibly and effectively.

### 3.5.2. Training

We use Adam [25] to optimize prediction loss $\mathcal{L}_{ours}(\theta)$. Adam algorithm performs one-step optimization for the stochastic objective function. The method computes individual adaptive learning rates for different parameters from the estimates of first and second moments of the gradients. The training process is aimed to find out the local minimum of $\mathcal{L}_{ours}(\theta)$, by using Adam to update the $\mathcal{L}_{ours}(\theta)$ w.r.t. parameters. The gradient of $\mathcal{L}_{ours}(\theta)$ concerning the parameters is present in Equation (15). The process of training in our work is performed in Algorithm 2. We empirically find out that the parameters have converged within 30 epochs.

$$\frac{\partial \mathcal{L}_{ours}\left(x_j^{(u)}, x_k^{(i)}, x'^{(u)}_j, x'^{(i)}_k \,|\theta\right)}{\partial \theta} = \sum_{(u,i) \in Q^+ \cup Q^-} (y_{ui} - \hat{y}_{ui})\left(\theta^T Z\right)' + \lambda \|\theta\| \tag{12}$$

$$\left(\theta^T Z\right)' = \begin{cases} \hat{y}^1_{ui}, & \theta = w_1 \\ \hat{y}^2_{u,i_{mix}}, & \theta = w_2 \\ w_1\left(\sum \widetilde{w}_i\right) + w_2\left(\sum \widetilde{w}_i + \left(\sum w'_u\right)\widetilde{w}'_{hi}\right), & \theta = (w_{hu})_{m \times |U|} \\ w_1\left(\sum w_u\right) + w_2\left(\sum w_u \widetilde{w}'_i(1 + \sum w'_u)\right), & \theta = (\widetilde{w}_{hi})_{m \times |I|} \\ w_2\left(\sum w_u \widetilde{w}'_i\left(\sum \widetilde{w}_i\right)\right), & \theta = (w'_{hu})_{m \times |U|} \\ w_2\left(\sum w_u\left(\sum w'_u \widetilde{w}_i\right)\right), & \theta = (\widetilde{w}'_{hi})_{m \times |I|} \end{cases}$$

where $w_u, w'_u \in R^m$ describe the column vectors of the user $u$ and selected user embedding weight matrix, respectively. Additionally, $w_{hu}$ and $w'_{hu}$ indicate their elements located in $h^{th}$; Likewise, $\widetilde{w}_i, \widetilde{w}'_i$ are the column of embedding weight matrix of the item $i$ and selected negative item and their elements $\widetilde{w}_{hi}, \widetilde{w}'_{hi}$, are in $h^{th}$.

---

**Algorithm 2:** Negative item mixing collaborative filtering.

**Input:** training set $\boldsymbol{u}, \boldsymbol{i}, \boldsymbol{u'}, \boldsymbol{i'}, \boldsymbol{y_{u,i}}; \boldsymbol{u'}, \boldsymbol{i'}$ are the selected negative samples

**Output:** parameters $\boldsymbol{\theta}$

1. Initialize $\boldsymbol{\theta}$
2. while not convergent do
3.     **for** all $\boldsymbol{u}, \boldsymbol{i}, \boldsymbol{u'}, \boldsymbol{i'}$ **do**
4.         calculate latent features $\boldsymbol{e_u}, \boldsymbol{e_i}, \boldsymbol{e_{u^-}}, \boldsymbol{e_{i^-}}$ from embedding layer
5.         calculate CF score $\boldsymbol{s(e_i, e_{u^-})}$ according to Equation (6)
6.         calculate predictor $\boldsymbol{\hat{y}_{u,i'}}$ according to Equations (7)–(9)
7.         perform parameters' updating via Adam algorithm
8.     **end** for
9. **end while**
10. **return** $\boldsymbol{\theta}$

---

## 4. Experiments

In this section, we present datasets and experiments to evaluate our proposed method. The algorithms were coded in Python 3.7.0, and computations were conducted on a personal computer with a Windows10 operating system, 2.3 GHz CPU, and 8 GB RAM. We aim to answer the following research questions:

RQ1: How does NMCF perform as compared to other state-of-the-art recommender system models?

RQ2: How do negative sample features affect NMCF?

RQ3: What are the effects of hyper-parameters on the NMCF model?

*4.1. Experimental Settings*

4.1.1. Dataset

We obtain the data from the Administrative Examination and Approval Bureau of Wuhou District, Chengdu. The dataset contains 106,630 interaction histories (14,129 users and 361 items) spanning from October 2015 to August 2019. Each user is identified by a unique user ID. The service items including public services, administrative examination, and approvals of the citizens, are identified by unique item ID. Table 1 exhibits the implicit interaction information of the dataset. Each entry includes the user ID, item ID, interaction time, and auxiliary information such as the user's gender (Male = 1, Female = 0). The data indicates users' behaviors, for example, a 54-year-old man with ID "001" clicked the item with ID "67540" on 21 October.

**Table 1.** Example of e-government service interaction entries.

| User ID | Gender | Age | Item ID | Time |
|---------|--------|-----|---------|------|
| 001 | 1 | 54 | 67540 | 21 October 2015 |
| 002 | 0 | 32 | 1453670 | 10 November 2015 |
| 003 | 1 | 30 | 12356 | 12 May 2017 |
| 004 | 0 | 27 | 67540 | 22 April 2018 |
| 005 | 0 | 45 | 3457091 | 10 July 2019 |

The recommendation system of personalized public service aims to predict a personalized top-K recommendation list for a user based on his interaction history. To simulate the real situation, we choose the leave-one-out scheme to split the train set and test set. For each user, we keep his latest action as the test set and train the remaining interaction histories. We follow the common strategy that randomly selects 100 negative items and then ranks the test item among the 100 negative items according to their predictive score [6,38]. Meanwhile, we also sample from the remaining items without user interaction for training. To avoid sample imbalance, we adopt the ratio between positive and negative samples (1:1).

4.1.2. Evaluation Metrics

We adopt the widely used metrics—including recall, normalized discounted cumulative gain (NDCG), precision, and F1—to evaluate the performance of personalized top-K item lists. For better understanding, Table 2 shows the evaluation metric for four users with three recommendation items. Specifically, the metric recall is the fraction of correctly predicted positive samples to the total positive samples. Thus, higher recall indicates a better performance. The formula is shown below as Equation (13).

$$Recall@K = \frac{1}{|U|}\sum_u \frac{TP}{TP + FN} \tag{13}$$

where, $|U|$ is the number of users, $TP$ and $FN$ are the number of true positive and false negative samples, respectively. True positive means the number of items in the top-N recommendation that hit the target. False negative means the number of the positive items test set that was falsely identified as the negative items.

Precision is the number of correctly predicted positive instances over the total number of predicted positive instances. Higher precision also indicates a better performance. See Equation (14) for details. Additionally, we use the F1 score to balance equally between Recall and Precision. The formula is shown in Equation (15)

$$Precision@K = \frac{1}{|U|}\sum_u \frac{TP}{TP + FP} = \frac{1}{|U|}\sum_u \frac{x}{K} \tag{14}$$

$$F1@K = \frac{2 * Precision@K * Recall@K}{(Precision@K + Recall@K)} \tag{15}$$

where *FP* is the number of false-positive instances. 'False positive' means the number of items in the top-N recommendation list that were falsely identified as the target items. The NDCG evaluates the gap between the predicted ranked item list and users' real interaction list. The closer the NDCG value is to 1, the better the model performance is.

$$NDCG@K = \frac{1}{|U|}\Sigma_u \frac{DCG@K}{IDCG@K} \tag{16}$$

$$DCG@K = \Sigma_{i=1}^K \frac{2^{r_i}-1}{\log_2(i+1)} \tag{17}$$

$$IDCG@K = \Sigma_{i=1}^{|REL|} \frac{2^{r_i}-1}{\log_2(i+1)} \tag{18}$$

where, $r_i = 1$ if the target test item in the position *i*; otherwise, $r_i = 0$; $|REL|$ indicates that the recommendation list is sorted in the best way. In the implicit feedback scenarios, $IDCG@K = 1$.

**Table 2.** Example evaluation metric for four users.

| User | Recommendation topK = 3 | Test Item | Recall@3 | Precision@3 | NDCG@3 |
|------|-------------------------|-----------|----------|-------------|--------|
| **User1** | [item1, item2, item3] | Item1 | 1/(1 + 0) | 1/3 | $(2-1)/log_2(2)$ |
| **User2** | [item2, item4, item5] | Item4 | 1/(1 + 0) | 1/3 | $(2-1)/log_2(3)$ |
| **User3** | [item1, iitem4, item3] | Item5 | 0/(0 + 1) | 0/3 | 0 |
| **User4** | [item3, itme4, item5] | Item5 | 1/(1 + 0) | 1/3 | $(2-1)/log_2(4)$ |
| **mean** | | | 0.75 | 0.25 | 0.53 |

### 4.1.3. Parameter Settings

We empirically select the optimal parameter settings for the proposed method. Specifically, we conduct a grid search to find out the optimal hyperparameters, which include the max number of epochs, learning rate, batch size, scale factors $\gamma_1$ and $\gamma_2$, and threshold $\varepsilon$. The size of the embedding layers is fixed to 64 for all the models, that is the dimension of latent features.

● Batch size

A batch is a set of samples that are randomly sampled from the training set. Its size is usually set to a larger value. We conduct experiments to search the optimal value of batch size in {128,256,521,1024}.

● Max number of the epoch

Epoch means one training over the entire sample dataset. Similarly, epoch is searched in {20,30,50,100}.

● Learning rate

The learning rate is the hyperparameter for updating weights in the process of gradient descent. Likewise, the learning rate is tested in {0.001,0.003,0.005,0.01}.

● Negative items selection parameters

For scale parameter $\gamma_1$ in the constructed metric function, we apply a grid search in {1,1.5,2,2.5,3}, for scale parameter $\gamma_2$, we apply a grid search in {0.02,0.04,0.06,0.08,0.10}. For distance constraint $\varepsilon$, we apply a grid search in {0.5,0.8,1.0,1.3,1.5}.

Consequently, the best values of batch size, max number of the epoch, learning rate, scale parameters $\gamma_1$ and $\gamma_2$, threshold $\varepsilon$ are determined: batch size = 1024, epoch = 30, learning rate = 0.003, $\gamma_1 = 2$ and $\gamma_2 = 0.06$, $\varepsilon = 1$, respectively.

#### 4.1.4. Comparative Methods

We compare our proposed model with the following methods:

• User-based CF (UCF)—UCF is the classical neighborhood-based collaborative filtering method. UCF obtains the new predictor scores by averaging (weighted) ratings over similar users. In other words, a user might choose according to ratings given to that item by the other users who have a similar preference with that of the target user. Similarity computation is critical to UCF. In this paper, we adopt the Pearson correlation coefficient because of its better performance. The Pearson correlation coefficient between users $u$ and $v$ is denoted as

$$pearson(u, v) = \frac{\sum_{i \in I} (r_{ui} - \bar{r}_u)(r_{vi} - \bar{r}_v)}{\sqrt{\sum_{i \in I} (r_{ui} - \bar{r}_u)^2} \sqrt{\sum_{i \in I} (r_{vi} - \bar{r}_v)^2}} \tag{19}$$

where, $I$ is the set of items.

• Baseline-MF—MF evaluates the observed user–item rating matrix by a linear combination of user and item latent features. It is one of the most successful realizations of latent features models. MF can be enforced by using the embedding neural network [39]. In our work, we apply an embedding layer to obtain the nonlinear latent features of baseline-MF.

As mentioned in Section 2, the inputs of the embedding layer are the one-hot vectors of users and items, and the outputs are the low-ranked vector representations. In this way, the users and items are embedded into the latent space with the help of the embedding layer. Finally, the MF used the dot of users and items to predict users' preferences on each item.

• Singular Value Decomposition (SVD)—SVD is a matrix factorization technique commonly used in a recommender system, producing low-rank approximations [27]. In the recommendation scenario, SVD is used to decompose the rating matrix $Q$ with the rank $r$ as

$$f_{SVD}(Q) = U \times S \times V^T \tag{20}$$

where, $U \in R^{|U| \times |U|}$ and $V \in R^{|I| \times |I|}$ are the orthogonal matrices, singular matrix $S \in R^{|U| \times |I|}$ is a diagonal matrix whose diagonal elements are non-negative real values. Especially, the rank $r'$ of the matrix $S$ is less than $r$. In this way, the rating matrix is transformed into a lower-ranked matrix. In this paper, we conduct an experiment for SVD on the e-government services dataset with the rank $r' = 64$.

• Time-Decayed BPR—On the basis of the Bayesian personalized ranking (BPR) framework, time-aware information is introduced [40]. Time-decayed BPR integrated the time-aware information and time-invariant information to model users' preferences. In this paper, the method was denoted as TBPR.

#### 4.2. Performance Comparison (RQ1)

In this part, we answer the first research question raised. For all results, we perform 10 technical replicates and take the mean as the final results. The experimental results of the top-5 are presented in Table 3. The detailed results (from top-1 to top-5) have been provided in Figure 5. We also use a pair-wise T-test to analyze all the results of the experiments. Through comparative analysis in this work come to the following conclusions:

(1) The proposed NMCF performed much better than MF, UCF, SVD, and TBPR in various evaluation metrics. The results demonstrate that NMCF is advantaged in the highly sparse e-government services dataset. The selected negative information successfully exploits the extra implicit preference of the users for better performance.

(2) Since the task of e-government recommendation is to find the target service item for users, we cared more about whether the target items have been selected in the top-K recommendation lists. Recall@5 score of the NMCF approach has shown that it hits more than 75% of users' target items in the top-5 recommendations, which has

improved by 51%, 41%, 18%, and 11% compared with that of SVD, UCF, TBPR, and MF, respectively.

(3) Since all experiments were performed 10 replicates, the freedom degree of t-distribution is 9. From Table 4, we accept the hypothesis that NMCF performs better than baseline models in terms of Recall@5, Precision@5, and NDCG@5 for significance levels of 0.0005 and 0.0025. The method NMCF successfully exploits the extra information, which is a high order indication of user preference, for a better recommendation.

**Table 3.** Results of top-5 evaluation metrics.

|  | Recall@5 | Precision@5 | F1@5 | NDCG@5 |
|---|---|---|---|---|
| UCF | 0.36 | 0.06 | 0.10 | 0.22 |
| Improve | 0.41 | 0.09 | 0.16 | 0.37 |
| SVD | 0.26 | 0.04 | 0.08 | 0.15 |
| Improve | 0.51 | 0.11 | 0.18 | 0.44 |
| TBPR | 0.59 | 0. 12 | 0.20 | 0.45 |
| Improve | 0.18 | 0.03 | 0.06 | 0.14 |
| MF | 0.66 | 0.13 | 0.22 | 0.46 |
| Improve | 0.11 | 0.02 | 0.06 | 0.13 |
| NMCF | 0.77 | 0.15 | 0.26 | 0.59 |

**Table 4.** *T*-test for paired comparisons.

|  | Recall@5 | | | | Precision@5 | | | | NDCG@5 | | | |
|---|---|---|---|---|---|---|---|---|---|---|---|---|
| **MF** | **UCF** | **SVD** | **TBPR** | **MF** | **UCF** | **SVD** | **TBPR** | **MF** | **UCF** | **SVD** | **TBPR** |
| **231.49** | 473.28 | 167.70 | 326.00 | 22.61 | 199.91 | 75.98 | 3.90 | 129.88 | 444.69 | 146.34 | 60.75 |
| **<0.0005** | <0.0005 | <0.0005 | <0.0005 | <0.0005 | <0.0005 | <0.0005 | <0.0025 | <0.0005 | <0.0005 | <0.0005 | <0.0005 |

### 4.3. Influence of Selection Negative Samples (RQ2)

In this section, we explore the influence of selection negative samples on NMCF. We focus on investigating the influence of threshold $\varepsilon$, which is the constraint for selecting controls. The evaluation scores at different thresholds are shown in Figure 6. The threshold follows the abscissa, the ordinate evaluation metrics score of our new model for result analysis. As we can see, the recall@1 is sensitive to the value of $\varepsilon$. When $\varepsilon = 1$, recall@1 of our model achieved the best result.

### 4.4. Sensitivity Analysis of Hyperparameters (RQ3)

In this part, we make a very thorough analysis of the influence of hyperparameters on the proposed method in the e-government dataset. Specifically, we investigate the impact hyperparameters of the learning rate and batch size. As can be seen from Table 5, the learning rate affects the performance of our model. This suggests that a lower learning rate might achieve a better result. Specifically, Recall@5, Precision@5, and F1@5 achieve the best results when the learning rate is 0.001. Next, we discuss the impact of batch size on NMCF. It can be seen from Table 6 that a lower batch size might achieve a better NDCG@5. Recall@5, precision@5, and F1@5 achieve the best result when the batch size is 1024.

**Table 5.** Impact of learning rate on the performance of NMCF.

| LR | Recall@5 | Precision@5 | F1@5 | NDCG@5 |
|---|---|---|---|---|
| 0.001 | 0.7768 | 0.1554 | 0.2589 | 0.5888 |
| 0.003 | 0.7731 | 0.1546 | 0.2577 | 0.59370 |
| 0.005 | 0.7616 | 0.1523 | 0.2539 | 0.5792 |
| 0.01 | 0.7482 | 0.14965 | 0.2494 | 0.5536 |

**Table 6.** Impact of batch size on the performance of NMCF.

| Batch Size | Recall@5 | Precision@5 | F1@5 | NDCG@5 |
|---|---|---|---|---|
| 128 | 0.7756 | 0.1551 | 0.2585 | 0.6019 |
| 256 | 0.7729 | 0.1546 | 0.2576 | 0.5922 |
| 512 | 0.7763 | 0.1553 | 0.2588 | 0.5965 |
| 1024 | 0.7768 | 0.1554 | 0.2589 | 0.5888 |

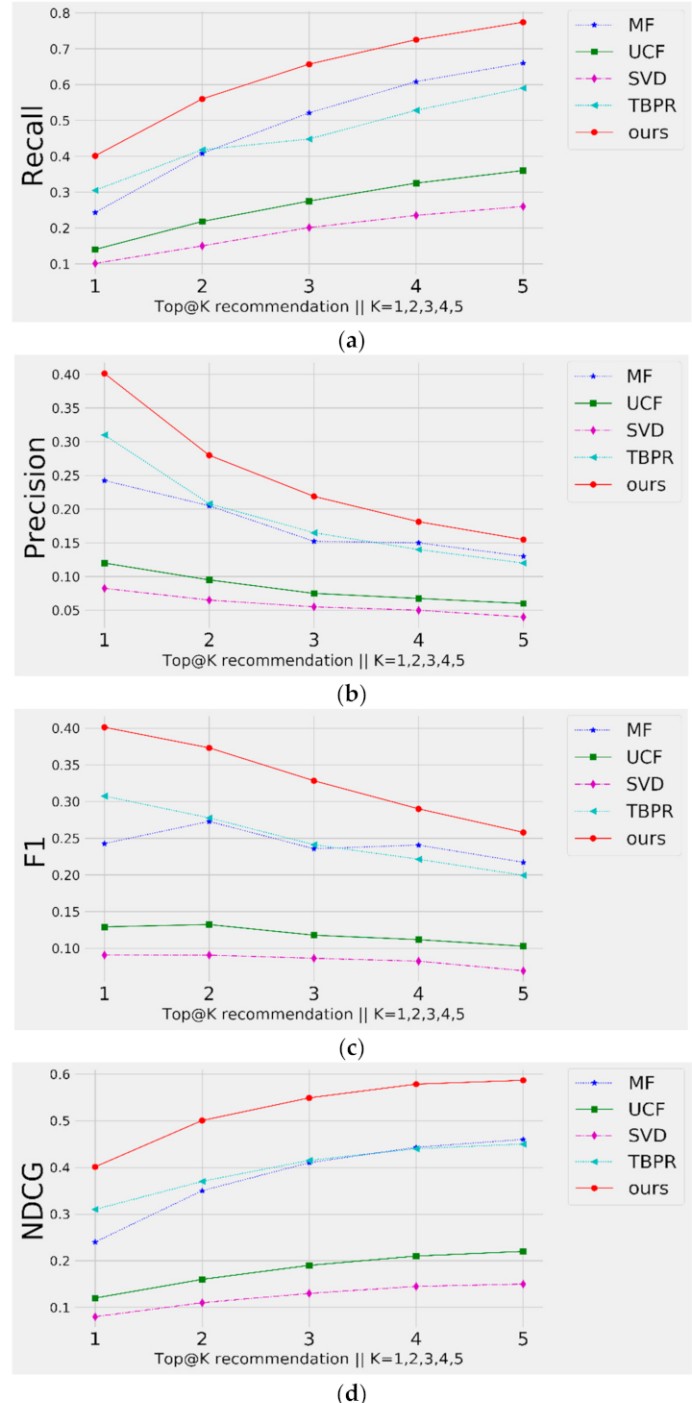

**Figure 5.** Comparison results (from top-1 to top-5) of different methods. (**a**–**d**) the results of NMCF compared with the baseline methods in terms of recall, precision, F1 score, and NDCG.

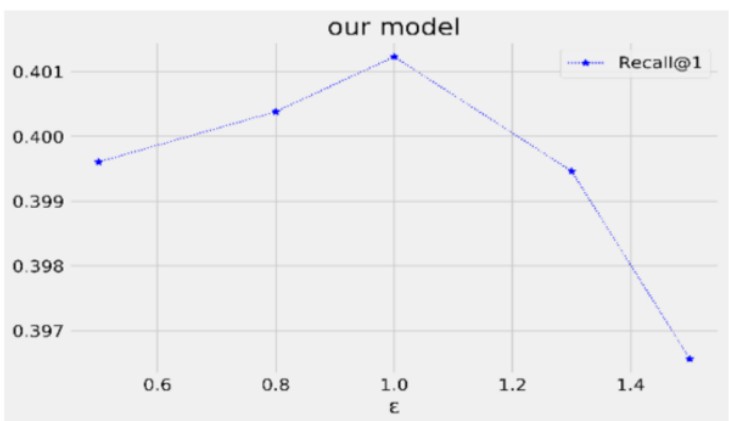

**Figure 6.** Selection of threshold $\varepsilon$ in our model.

## 5. Conclusions and Future Work

In this paper, the aim is to capture more latent features to model users' preferences. We presented an efficient NMCF method for e-government services recommendations. Different from evaluating the similarity of users or items (such as neighborhood-based CF), NMCF uses the implicit characteristics (e.g., interaction timestamp) and users' ages to evaluate the distance between the user and the item interaction. In this study, NMCF not only learns the interaction information but also capture feature from negative items based on the similarity of interaction pairs. The proposed model is closely related to real e-government problem settings and can be rapidly reused in practice without feature engineering. It can effectively solve the problems encountered in the e-government platforms, such as lower willingness to use because of bad user experience. For example, users may be unable to transform spoken language (such as 'baby') into written language (such as 'neonates'), let alone find the items regard neonates. Thus, our method can effectively solve the problem by generating personalized recommendations for users. Two potential limitations need to be considered. Firstly, the similarity measure constructed in this paper may consume more time, when the number of users and items is huge. Secondly, the randomly selected items based on a similarity metric might not adequately capture users' preferences. In future work, we will put more effort into mining users' fine-grained preferences based on attribute information with the help of an attention mechanism. In this way, we can not only capture users' fine-grained preferences for items but also improve the performance of the model.

**Author Contributions:** Conceptualization, T.C.; Methodology, N.S.; Validation, T.C., N.S. and W.G.; Resources, T.C.; Data curation, N.S.; Writing—original draft preparation, N.S.; Writing—review and editing, T.C. and L.R.; Project administration, W.G. and L.R.; Funding acquisition, T.C. and N.S. All authors have read and agreed to the published version of the manuscript.

**Funding:** The APC was funded by (1) Fundamental Research Funds for the Central Universities, China No. HUST: 2020JYCXJJ036; (2) Humanities and Social Science Fund of Ministry of Education of China No.19YJA630010; (3) National Natural Science Foundation of China No.71734002, 72042016; and (4) Chinese National Funding of Social Sciences No.18ZDA109, 17ZDA102.

**Institutional Review Board Statement:** Not applicable.

**Informed Consent Statement:** Not applicable.

**Data Availability Statement:** Not applicable.

**Acknowledgments:** We would like to thank the Administrative Examination and Approval Bureau of Wuhou District, Chengdu (Wuhou District Government Affairs Service Center) for providing us with the dataset.

**Conflicts of Interest:** The authors declare no conflict of interest.

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
