# Peer review of "Enhanced Collaborative Filtering for Personalized E-Government Recommendation"

_applsci, doi:10.3390/app112412119_

Round 1
Reviewer 1 Report
This paper presents a negative items mixed collaborative filtering method (NMCF) to enhance the collaborative filtering-based recommender system.
Pros.
- The proposed method is well motivated. The authors exploit mixing negative information to the personalized e-gov service recommendation.
- The authors design an effective time-sensitive distance metric for recommendation systems.
- The proposed method is evaluated on real-world datasets.
Cons.
- The paper presentation should be improved. In some sections, the formats are messed, e.g., titles of Section 3.1 and Section 3.2 are not aligned.
- Experimental part could be improved by adding hyperparameter analysis.
- Collaborative filtering and user modeling are very active research topics. Some additional references include:
[a] A Survey on Representation Learning for User Modeling. The 29th International Joint Conference on Artificial Intelligence, 2020.
[b] Deep learning based recommender system: A survey and new perspectives. ACM CSUR, 2019.
[c] Deep Collaborative Filtering via Marginalized Denoising Auto-encoder. The 24th ACM International Conference on Information and Knowledge Management, 2015.
Author Response
Point 1: The paper presentation should be improved. In some sections, the formats are messed, e.g., titles of Section 3.1 and Section 3.2 are not aligned.
Response 1: Thank you for taking the time to review the manuscript. Following your feedback and suggestions, in our newly submitted manuscript, we proofread the format carefully and improved the presentation. We sincerely hope that these changes have improved the paper.
Point 2: Experimental part could be improved by adding hyperparameter analysis.
Response 2: Thank you for the comment. Following your feedback and suggestions, we elaborated on the research questions at the beginning of Section 4. In Section 4.3 and 4.4, we conduct the sensitivity analysis of hyperparameters. And, the results are presented in Table 5, Table 6, and Figure 6. Specifically, we explained the impact of the learning rate, batch size, and captured negative sample features.
Point 3: Collaborative filtering and user modeling are very active research topics. Some additional references include:
[a] A Survey on Representation Learning for User Modeling. The 29th International Joint Conference on Artificial Intelligence, 2020.
[b] Deep learning based recommender system: A survey and new perspectives. ACM CSUR, 2019.
[c] Deep Collaborative Filtering via Marginalized Denoising Auto-encoder. The 24th ACM International Conference on Information and Knowledge Management, 2015.
Response 3: Thank you for pointing this out. Following your feedback and suggestions, we have added references in Lines 154-168.
Reviewer 2 Report
The paper contains sufficiently new and adequate information, and it adheres to the journal’s standards. The topic and level of formality are appropriate for the journal`s readership. Its style and readability are suitable. There is a huge amount of information given throughout the article, but I would suggest revising the paper.
The methodological concept is clear. The selected methodology is scientifically appropriate, but the methods are not described in detail. I suggest to describe better the Evaluate metrics and Comparative methods.
I also miss recent relevant literature in this area. I suggest citing: TOMAŽIČ, Tina, UDIR MIŠIČ, Katja. Parliament-citizen communication in terms of local self-government and their use of social media in the European Union. Lex localis. Oct. 2019, vol. 17, no. 4, 1057-1079.
Results are presented clearly and analyzed appropriately. Major idea received enough attention and explanation.
I recommend rewriting the conclusion. The concluding remarsk should be more specific and better explained. The further study is mentioned, but it should be more concretized.
In summary, the article is sufficiently interesting to warrant publication, but it needs major revision. Please follow all the comments above.
Author Response
Point 1: The methodological concept is clear. The selected methodology is scientifically appropriate, but the methods are not described in detail. I suggest to describe better the Evaluate metrics and Comparative methods.
Response 1: Thank you for taking the time to review the manuscript. Following your feedback and suggestions, in our newly submitted manuscript, we explain the Evaluate metrics in detail. In addition, we rewrite the Comparative methods in Section 4.1. We sincerely hope that these changes have improved the paper.
Point 2: I also miss recent relevant literature in this area. I suggest citing: TOMAŽIČ, Tina, UDIR MIŠIČ, Katja. Parliament-citizen communication in terms of local self-government and their use of social media in the European Union. Lex localis. Oct. 2019, vol. 17, no. 4, 1057-1079. AI-based systems.
Response 2: Thank you for the comment. Following your feedback and suggestions, we have added the reference in our work. We sincerely hope that these changes have improved the paper.
Point 3: I recommend rewriting the conclusion. The concluding remarks should be more specific and better explained. The further study is mentioned, but it should be more concretized.
Response 3: Thank you for the comment. Following your feedback and suggestions, we rewriting the conclusion. And, we detailed the further study in the conclusion. We sincerely hope that these changes have improved the paper.
Reviewer 3 Report
The paper proposes using an intelligent recommendation system on e-government platforms. The paper needs to be improved to include a better description of methodology and experiments as well as supported with statistical analysis of the results.
Comments:
- The authors should clearly state the novelty of their approach and its difference from multiple other collaboration filtering methods known from the literature. The Contribution paragraph on page 2 rather summarizes all the works done and presented in this study rather than stating actual contributions to the research field. For example, item 5 of contributions is more fitting for the discussion or conclusions section. Therefore, this part of the paper needs to be rewritten.
- The related works section discusses some recommendation systems and approaches using some traditional methods in general. I suggest to supplement it with a discussion of hybrid method, which gained some popularity recently, such as “Recommendation based on review texts and social communities: A hybrid model”, “Hybrid neural recommendation with joint deep representation learning of ratings and reviews”, and “Intelligent recommendation of related items based on naive bayes and collaborative filtering combination model.”
- I could not find Algorithm 2.
- Provide a link to the dataset used in this study.
- Where is the positive/negative interaction label in your dataset (Table 1)?
- Equations 13-18: explain all variables and notations used.
- Is the improvement of results statistically significant? Perform statistical analysis to support.
- Figure 6: improve size and text label size for readability. Explain each sub-figure in the caption of the figure.
- Discuss in more detail the limitations of the proposed method and threats-to-validity of the experimental results. Lines 88-89: “It can effectively solve the problems encountered in the e-government platforms” -> discuss specifically (with examples) what problems your approach could solve. Adding a real-world case study would benefit the article.
- Improve the conclusions. Go beyond a simple summary of the works done.
Author Response
Point 1: The authors should clearly state the novelty of their approach and its difference from multiple other collaboration filtering methods known from the literature. The Contribution paragraph on page 2 rather summarizes all the works done and presented in this study rather than stating actual contributions to the research field. For example, item 5 of contributions is more fitting for the discussion or conclusions section. Therefore, this part of the paper needs to be rewritten.
Response 1: Thank you for taking the time to review the manuscript. Following your feedback and suggestions, in our newly submitted manuscript, we detailed the novelty and difference of our model from other methods. And, rewrite the contribution paragraph on page 2. We sincerely hope that these changes have improved the paper.
Point 2: The related works section discusses some recommendation systems and approaches using some traditional methods in general. I suggest to supplement it with a discussion of hybrid method, which gained some popularity recently, such as “Recommendation based on review texts and social communities: A hybrid model”, “Hybrid neural recommendation with joint deep representation learning of ratings and reviews”, and “Intelligent recommendation of related items based on naive bayes and collaborative filtering combination model.”
Response 2: Thank you for the comment. Following your feedback and suggestions, we have added references. We sincerely hope that these changes have improved the paper.
Point 3: I could not find Algorithm 2.
Response 3: Thank you for the comment. Following your feedback and suggestions, we have added Algorithm 2 in line 293. We sincerely hope that these changes have improved the paper.
Point 4: Provide a link to the dataset used in this study.
Response 4: Thank you for the suggestion. We’ve discussed this issue with our government partners however they are unlikely to share the data. Sorry about this.
Point 5: Where is the positive/negative interaction label in your dataset (Table 1)?
Response 5: Thank you for the comment. As for your question, we have explained the data we collected in the government service hall in Wuhou District in Section 4.1. We sincerely hope that these changes have improved the paper.
Point 6: Equations 13-18: explain all variables and notations used.
Response 6: Thank you for the comment. Following your feedback and suggestions, in Equations 13-18, we add the explanation in detail. We sincerely hope that these changes have improved the paper.
Point 7: Is the improvement of results statistically significant? Perform statistical analysis to support.
Response 7: Thank you for the comment. Following your feedback and suggestions, we have added the pair-wise T-test to analyze all the results. The results are shown in Table 4. We sincerely hope that these changes have improved the paper.
Point 8: Figure 6: improve size and text label size for readability. Explain each sub-figure in the caption of the figure.
Response 8: Thank you for taking the time to review the manuscript. Following your feedback and suggestions, in our newly submitted manuscript, we modified the figure in Section 4.2. We sincerely hope that these changes have improved the paper.
Point 9: Discuss in more detail the limitations of the proposed method and threats-to-validity of the experimental results. Lines 88-89: “It can effectively solve the problems encountered in the e-government platforms” -> discuss specifically (with examples) what problems your approach could solve. Adding a real-world case study would benefit the article.
Response 9: Thank you for the comment. Following your feedback and suggestions, we have discussed the limitation of the model in Section 5. Additionally, we also exemplify the difficulty that users may encounter when using the government platform. We sincerely hope that these changes have improved the paper.
Point 10: Improve the conclusions. Go beyond a simple summary of the works done.
Response 10: Thank you for the comment. Following your feedback and suggestions, we rewrite the conclusion. Specifically, we summarize the method, describe the limitations, and explain the future work in detail. We sincerely hope that these changes have improved the paper.
Round 2
Reviewer 2 Report
Dear authors, I think you revised all the necessary comments and I suggest to publish the manuscript in the present form.
Kind regards!
Reviewer 3 Report
The article has been revised well. I recommend to accept.